# Upscaling of Latent Heat Flux in Heihe River Basin Based on Transfer Learning Model

Jing Lin [1], Tongren Xu [1,*], Gangqiang Zhang [1], Xiangping He [1], Shaomin Liu [1], Ziwei Xu [1], Lifang Zhao [2], Zongbin Xu [1] and Jiancheng Wang [1]

1   State Key Laboratory of Earth Surface Processes and Resource Ecology, School of Natural Resource, Faculty of Geographical Science, Beijing Normal University, Beijing 100875, China
2   Aerospace Information Research Institute, Chinese Academy of Sciences, Beijing 100094, China
*   Correspondence: xutr@bnu.edu.cn

**Abstract:** Latent heat flux (LE) plays an essential role in the hydrological cycle, surface energy balance, and climate change, but the spatial resolution of site-scale LE extremely limits its application potential over a regional scale. To overcome the limitation, five transfer learning models were constructed based on artificial neural networks (ANNs), random forests (RFs), extreme gradient boosting (XGBoost), support vector machine (SVM), and light gradient boosting machine (LightGBM) to upscale LE from site scale to regional scale in Heihe River basin (HRB). The instance-transfer approach that utilizes data samples outside of HRB was used in the transfer learning models. Moreover, the Bayesian-based three-cornered hat (BTCH) method was used to fuse the best three upscaling results from ANN, RF, and XGBoost models to improve the accuracy of the results. The results indicated that the transfer learning models perform best when the transfer ratio (the data samples ratio between external and HRB dataset) was 0.6. Specifically, the coefficient of determination ($R^2$) and root mean squared errors (RMSE) of LE upscaled by ANN model was improved or reduced by 6% or 17% than the model without external data. Furthermore, the BTCH method can effectively improve the performance of single transfer learning model with the highest accuracy ($R^2 = 0.83$, RMSE = 18.84 W/m$^2$). Finally, the LE upscaling model based on transfer learning model demonstrated great potential in HRB, which may be applicable to similar research in other regions.

**Keywords:** artificial neural networks; random forests; extreme gradient boosting; transfer learning; product ensemble

## 1. Introduction

Latent heat flux (LE) or evapotranspiration (ET) is an important process of hydrological cycle and surface energy balance [1–3], including soil and open water evaporation, plant transpiration, and canopy interception evaporation. LE is vital for climate change, which accounts for about 60% of global precipitation [4]. As irrigation water use increases, so does competition for water across regions. The basin-scale LE plays a crucial role in directing the efficient distribution and administration of water resources.

Currently, the methods of obtaining LE mainly include lysimeters, eddy covariance (EC) systems, large aperture scintillometers (LASs) and optical-microwave scintillometers (OMSs) at the site scale or field scale [5–7]. The regional-scale LE can be acquired through remote sensing estimation methods such as empirical statistical method [8,9], energy balance method [10–12], and variational data assimilation method [13–15]. However, most of the methods are still only applicable to small scales, and some errors will occur when the study area is extended to watershed or global scale. In addition, many parameters in the models cannot be measured directly due to the limitations of ground-based observations unless they are obtained by empirical estimation or parameter optimization, which will result in a strong geographical variability of the final calculated LE.

To acquire the accurate regional LE, many researchers have made numerous attempts at upscaling approaches based on ground-measured observations. The common upscaling methods include four major groupings, and the detailed descriptions are as follow. The first group averages the values of the ground sampling points directly [16] or weights them according to the area or footprint range [17,18]. It is easy and effective to obtain the regional information from site data through these methods, but limited by the spatial heterogeneity and reasonable sampling strategy. Establishing empirical regression models is another widely used method, which involves creating a correlation between land surface variables and the target variable based on site-scale datasets, and subsequently extrapolating the relationship to a basin-scale level [19]. Nevertheless, the empirical regression method, although useful for small areas, may not be suitable for extrapolation to larger areas. The third group, called "geostatistical methods", mainly studies the natural phenomena with spatial correlation based on kriging theoretical framework or Bayesian theory framework, and it is widely used in the study of spatial scale expansion from point scale to regional scale [20,21]. The fourth group is based on machine learning techniques such as artificial neural networks [22], random forests [23], and so on [24–27]. However, what is captured by machine learning is limited to the training set that is input to the model. As a result, predictions in ranges outside the training set are not as effective, especially for extreme weather [28].

Although previous methods have elevated the site-scale LE to regional scale to some extent, these methods have mostly considered data within regions or near sites but not in other similar regions, and the advent of transfer learning provides a new perspective. Transfer learning is inspired by the ability to transfer knowledge across domains and aims to use knowledge from the source domain to improve the learning performance of the target domain or to reduce the number of labeled examples required for the target domain [29]. It can be divided into instance transfer, feature-representation transfer, parameter transfer, and relational-knowledge transfer based on the content of the transfer [30]. Instance transfer means that the dataset of the source domain can be reused together with the data of the target domain after corresponding processing, and has a wide range of applications due to its relatively simple principle. Wu and Dieterich [31] integrated source domain data into the support vector machine framework to improve classification performance. Feature-representation transfer aims to find the good feature representation and minimize the error of classification or regression model in the target domain. Parameter transfer is essentially the migration of parameters or weights from the source task to the target task to save the training time of the model. As the weights of the loss functions may vary, assigning larger weights to the loss functions in the target domain ensures better performance in the target domain. Lawrence and Platt [32] proposed an efficient method for learning the parameters of a Gaussian process (GP), called multi-task informative vector machine (MT-IVM), for handling multi-task learning situations. Schwaighofer and Kai Yu [33] combined hierarchical Bayesian (HB) with GP for multi-task learning. Unlike the above three transfer learning methods, the relational-knowledge transfer deals with the transfer learning problems in the relational domain and involves the non-independent and identically distributed data. Mihalkova et al. [34] proposed transfer via automatic mapping and revision (TAMAR), which uses Markov logic networks (MLNs) to transfer relational knowledge across relational domains. Zhao et al. [35] introduced transfer learning into the field of natural hazards, providing a new idea for improving machine learning-based assessment methods by transferring prior knowledge of different catchments.

Sometimes, the capability of an individual transfer learning model is limited due to the limitation of the number of sites and land cover types, in which case the fusion of multiple models seems to solve this problem. In contrast to other fusion methods [36,37], the Bayesian-based three-cornered hat (BTCH) approach enables the integration of products without using any a priori knowledge, and optimizes the use of available information, ultimately enhancing the quality and accuracy of data fusion. Given the robustness of machine learning models and the effectiveness of fusion algorithm, this study investigated

the performance of five transfer learning methods and the BTCH fusion method based on the observed data from the Heihe River basin (HRB) sites and similar external datasets. Five machine learning algorithms are employed in this research, which are artificial neural networks (ANNs), random forests (RFs), support vector machine (SVM), extreme gradient boosting (XGBoost, version 1.3.1, Python 3.6), and light gradient boosting machine (Light-GBM), respectively, and the best three of them are selected for fusion to obtain the optimal upscaled daily LE with the spatial resolution of 1 km in 2018.

## 2. Materials

### 2.1. Study Area

The HRB, located in Northwest China, has a dry and cold climate typical of a temperate continental monsoon region. In the upstream of HRB, the climate is wet and cold, with annual-averaged air temperatures below 2 °C, and annual precipitation is about 350 mm. The main land cover types include forest (mainly Qinghai spruce), grassland (alpine meadow), etc. The midstream region is located in Hexi Corridor, where annual precipitation decreases from 250 mm in the south to less than 100 mm in the north, mainly covered by oases, oasis–desert transition zone, desert grassland, and desert. Most of the oases in the midstream are artificial oases covered with maize. The desert species are scarce but drought-tolerant, which are spatially distributed in patchy patterns. The downstream is extremely dry, with annual precipitation less than 50 mm, and the dominant landscapes are desert and natural oases (Tamarix and Populus Euphratica) (Figure 1).

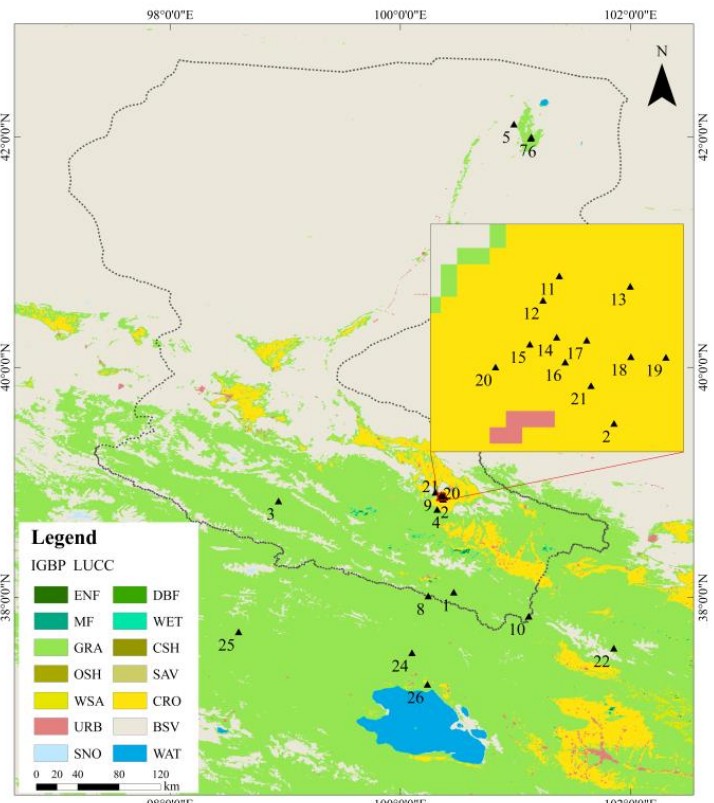

**Figure 1.** Land cover and flux tower site locations of the HRB. ENF—evergreen needleleaf forest; DBF—deciduous broadleaf forest; MF—mixed forest; WET—permanent wetlands; GRA—grasslands; CSH—closed shrublands; OSH—open shrublands; SAV—savannas; WSA—woody savannas; CRO—croplands; URB—urban and built-up; BSV—barren or sparsely vegetated; SNO—snow and ice; WAT—water bodies.

### 2.2. Data in Heihe River Basin

The dataset of hydrometeorological observation network in Heihe Watershed allied telemetry experimental research (HiWATER), the dataset of Heihe integrated observatory network in Qilian Mountains integrated observatory network, and the dataset of flux observation matrix in the multi-scale observation experiment on evapotranspiration over heterogeneous land surfaces (HiWATER-MUSOEXE) are used in this study (https://data.tpdc.ac.cn/home, accessed on 1 February 2023). LE and meteorological observations were obtained at 21 stations for model validation at site, including 3 barren or sparsely vegetated (BSV) sites, 12 cropland (CRO) sites, 1 closed shrubland (CSH) site, 1 deciduous broadleaf forest (DBF) site, and 4 grassland (GRA) sites. The specific information of the above sites was summarized in Table 1 [5,7,17,38–40], and the distribution can be found in Figure 1. The upstream sites basically include the main vegetation types in the upstream of HRB, namely alpine meadow and Qinghai spruce. Sites in the midstream are primarily covered by cropland, and intensive observation was carried out in the oases from June to September 2012. The downstream sites are densely distributed near the oases.

**Table 1.** Summary of eddy covariance flux tower sites in HRB.

| NO. | Site Name | Longitude (W°) | Latitude (N°) | Land Cover | Period |
|---|---|---|---|---|---|
| 1 | Arou | 100.4643 | 38.0473 | GRA | 2013/1–2019/12 |
| 2 | Daman | 100.3722 | 38.8555 | CRO | 2013/1–2019/12 |
| 3 | Dashalong | 98.9406 | 38.8399 | GRA | 2013/8–2019/12 |
| 4 | Huazhaizi desert steppe | 100.3186 | 38.7652 | BSV | 2016/1–2017/12 |
| 5 | Desert | 100.9872 | 42.1135 | BSV | 2016/1–2019/12 |
| 6 | Mixed forest | 101.1335 | 41.9903 | DBF | 2013/7–2019/12 |
| 7 | Sidaoqiao | 101.1374 | 42.0012 | CSH | 2016/1–2019/12 |
| 8 | Yakou | 100.2421 | 38.0142 | GRA | 2016/1–2019/12 |
| 9 | Bajitan Gobi | 100.3042 | 38.9150 | BSV | 2012/8–2015/4 |
| 10 | Jingyangling | 101.1160 | 37.8384 | GRA | 2018/1–2019/12 |
| 11 | Ponit 1 | 100.3582 | 38.8932 | CRO | 2012/6–2012/9 |
| 12 | Ponit 2 | 100.3541 | 38.8870 | CRO | 2012/6–2012/9 |
| 13 | Ponit 3 | 100.3763 | 38.8905 | CRO | 2012/6–2012/9 |
| 14 | Ponit 4 | 100.3575 | 38.8775 | CRO | 2012/6–2012/9 |
| 15 | Ponit 5 | 100.3507 | 38.8757 | CRO | 2012/6–2012/9 |
| 16 | Ponit 6 | 100.3597 | 38.8712 | CRO | 2012/6–2012/9 |
| 17 | Ponit 7 | 100.3652 | 38.8768 | CRO | 2012/6–2012/9 |
| 18 | Ponit 8 | 100.3765 | 38.8725 | CRO | 2012/6–2012/9 |
| 19 | Ponit 9 | 100.3855 | 38.8724 | CRO | 2012/6–2012/9 |
| 20 | Ponit 11 | 100.3420 | 38.8699 | CRO | 2012/6–2012/9 |
| 21 | Ponit 12 | 100.3663 | 38.8652 | CRO | 2012/6–2012/9 |

Site-scale LE can be observed by eddy covariance systems, and the raw observations were 10 Hz data, which have been post-processed by EddyPro software developed by Li-Cor for obtaining 30 min flux data (http://www.licor.com/env/products/eddy_covariance/software.html, accessed on 1 February 2023). In addition, the study preprocessed the data accordingly: (1) The half-hour flux data were aggregated to daily scale; (2) Bowen ratio correction was used to correct energy balance closure of the heat flux [41].

The meteorological variables were observed synchronously by automatic weather stations, including air temperature (Ta), humidity (RH), wind speed (WS), atmospheric pressure (Pa), precipitation (P), vapor pressure deficit (VPD), soil moisture (SM), and net radiation (Rn). The daily automatic weather station data were calculated by averaging 10 min raw data, and the precipitation was accumulated for 30 days to minimize the effect of canopy interception.

LE obtained from LAS observations at the three stations of Arou, Daman, and Sidaoqiao during the HiWATER and HiWATER-MUSOEXE experiments in 2018 were also used as independent validation data for the upscaled results [5,38,39]. LAS express the turbulence

intensity of the atmosphere by measuring the structural parameters of the air refraction index, which in turn derives the sensible heat flux (H), and calculates LE using the surface energy balance residual method by combining Rn and soil heat fluxes (G) observed by automatic weather stations.

### 2.3. External Data of Heihe River Basin

In our study, in addition to selecting relevant stations within HRB as training data for the model, observation sites outside the basin were also selected to expand the training set.

Five flux sites located in the upstream of HRB (Figure 1), namely Xiyinghe site, Liancheng site, alpine meadow and grassland site, subalpine shrub site, and temperate steppe site, were collected from the dataset of Qinghai Lake integrated observatory network in Qilian Mountains integrated observatory network and the cold and arid research network of Lanzhou University (CARN, https://data.tpdc.ac.cn/home, accessed on 1 February 2023). The land cover types of the above stations are mainly GRA and CSH (Table 2) [42–49].

**Table 2.** Information of the sites in Qilian Mountains integrated observatory network.

| NO. | Site Name | Longitude (W°) | Latitude (N°) | Land Cover | Period |
|-----|-----------|----------------|---------------|------------|--------|
| 22 | Xiyinghe | 101.8550 | 37.5610 | GRA | 2019–2020 |
| 23 | Liancheng | 102.7370 | 36.6920 | GRA | 2019 |
| 24 | Subalpine shrub | 100.1010 | 37.5210 | CSH | 2019–2020 |
| 25 | Alpine meadow and grassland | 98.5949 | 37.7032 | GRA | 2018–2020 |
| 26 | Temperate steppe | 100.2358 | 37.2469 | GRA | 2019–2020 |

The FLUXNET 2015 dataset (https://fluxnet.org/data/fluxnet2015-dataset, accessed on 5 February 2023) was also used to supplement the training set. A total of 37 sites were extracted from the dataset based on the availability of variables and the similar land cover types as the HRB sites, including 13 CRO sites, 23 GRA sites, and 1 CSH site (Table 3).

**Table 3.** Information of the sites in FLUXNET 2015 dataset.

| NO. | Site Name | Latitude (N°) | Longitude (W°) | Land Cover | Period |
|-----|-----------|---------------|----------------|------------|--------|
| 1 | AU-DaP | −14.0633 | 131.3181 | GRA | 2007–2013 |
| 2 | AU-Emr | −23.8587 | 148.4746 | GRA | 2011–2013 |
| 3 | AU-Rig | −36.6499 | 145.5759 | GRA | 2011–2014 |
| 4 | AU-Stp | −17.1507 | 133.3502 | GRA | 2008–2014 |
| 5 | AU-TTE | −22.2870 | 133.6400 | GRA | 2012–2014 |
| 6 | AU-Ync | −34.9893 | 146.2907 | GRA | 2012–2014 |
| 7 | BE-Lon | 50.5516 | 4.7462 | CRO | 2004–2014 |
| 8 | CH-Cha | 47.2102 | 8.4104 | GRA | 2006–2012 |
| 9 | CH-Oe1 | 47.2858 | 7.7319 | GRA | 2003–2008 |
| 10 | CN-Cng | 44.5934 | 123.5092 | GRA | 2007–2010 |
| 11 | CN-Du2 | 42.0467 | 116.2836 | GRA | 2007–2008 |
| 12 | CN-Du3 | 42.0551 | 116.2809 | GRA | 2009–2010 |
| 13 | CN-HaM | 37.3700 | 101.1800 | GRA | 2002–2004 |
| 14 | CZ-BK2 | 49.4944 | 18.5429 | GRA | 2006–2012 |
| 15 | DE-Geb | 51.0997 | 10.9146 | CRO | 2001–2011 |
| 16 | DE-Kli | 50.8931 | 13.5224 | CRO | 2004–2014 |
| 17 | DE-Seh | 50.8706 | 6.4497 | CRO | 2007–2010 |
| 18 | DK-Fou | 56.4842 | 9.5872 | CRO | 2005 |
| 19 | FR-Gri | 48.8442 | 1.9519 | CRO | 2005–2012 |
| 20 | IT-BCi | 40.5237 | 14.9574 | CRO | 2004–2014 |
| 21 | IT-CA2 | 42.3772 | 12.0260 | CRO | 2011–2012 |

**Table 3.** *Cont.*

| NO. | Site Name | Latitude (N°) | Longitude (W°) | Land Cover | Period |
|-----|-----------|---------------|----------------|------------|--------|
| 22 | IT-MBo | 46.0147 | 11.0458 | GRA | 2004–2013 |
| 23 | PA-SPs | 9.3138 | −79.6314 | GRA | 2007–2009 |
| 24 | RU-Ha1 | 54.7252 | 90.0022 | GRA | 2002–2004 |
| 25 | US-AR1 | 36.4267 | −99.4200 | GRA | 2009 |
| 26 | US-AR2 | 36.6358 | −99.5975 | GRA | 2009 |
| 27 | US-ARb | 35.5497 | −98.0402 | GRA | 2005–2006 |
| 28 | US-ARc | 35.5465 | −98.0400 | GRA | 2005–2006 |
| 29 | US-ARM | 36.6058 | −97.4888 | CRO | 2003–2012 |
| 30 | US-CRT | 41.6285 | −83.3471 | CRO | 2011–2013 |
| 31 | US-KS2 | 28.6086 | −80.6715 | CSH | 2003–2006 |
| 32 | US-Lin | 36.3566 | −119.0922 | CRO | 2009 |
| 33 | US-SRG | 31.7894 | −110.8277 | GRA | 2008–2014 |
| 34 | US-Tw2 | 38.0969 | −121.6365 | CRO | 2012–2013 |
| 35 | US-Tw3 | 38.1152 | −121.6469 | CRO | 2013–2014 |
| 36 | US-Var | 38.4133 | −120.9508 | GRA | 2000–2014 |
| 37 | US-Wkg | 31.7365 | −109.9419 | GRA | 2004–2014 |

### 2.4. Meteorological and Remote Sensing Forcing Data

In conjunction with site-scale observations, corresponding meteorological and remote sensing data were collected to enrich the features of upscaling model, and the detailed information can be found in Table 4. The meteorological data used in this study is a kind of atmospheric forcing data in the HRB (2000–2021), which is produced based on the weather research and forecasting (WRF) model. Based on this dataset, this study acquired 5 variables (Ta, Pa, P, RH, and WS) with the temporal/spatial resolution of 1 h/0.05° [50–52]. Additionally, SM dataset was produced by the daily 0.05° × 0.05° land surface soil moisture dataset of the Qilian Mountain area (2018, SMHiRes, V1) [53,54] and Rn dataset was obtained from the simulated forcing dataset of 3 km/6 h in HRB (1980–2080) [55].

**Table 4.** Input datasets used to drive the machine learning model.

| Variable | Dataset | Spatial Resolution | Temporal Resolution |
|----------|---------|--------------------|--------------------|
| Ta/Pa/P/RH/WS | The atmospheric forcing data (2000–2021) | 0.05° × 0.05° | Hourly |
| SM | 2018, SMHiRes, V1 | 0.05° × 0.05° | Daily |
| Rn | Simulated forcing dataset (1980–2080) | 3 km × 3 km | 6 h |
| LAI | MCD15A2H | 500 m × 500 m | 8 days |
| NDVI | MOD13A2 | 1 km × 1 km | 16 days |
| LC | MCD12Q1 | 500 m × 500 m | Yearly |
| DEM | SRTMDEM | 90 m × 90 m | - |
| SLOPE | SRTMSLOPE | 90 m × 90 m | - |
| ASPECT | SRTMASPECT | 90 m × 90 m | - |

The remote sensing datasets used as input in this study include leaf area index (LAI), normalized difference vegetation index (NDVI), and land cover (LC) (https://modis.gsfc.nasa.gov, accessed on 7 February 2023). LAI and NDVI data, with a time resolution of 8 days and 16 days, respectively, were linearly interpolated to daily scale and were also extracted the values at sites as a complement to the driving variables for the transfer learning model. The MCD12Q1 product was used in this study to delineate the land cover type of HRB. Since the selected model cannot directly recognize textual variables, One-Hot Encoding is introduced to process land cover type data.

The Food and Agriculture Organization (FAO) 56 Penman–Monteith (P-M) equation (see Equation (1)) was also used to calculate reference crop evapotranspiration ($ET_0$) at the site and within the watershed as one of the driving variables for the machine learning model.

$$ET_0 = \frac{0.408\Delta(R_n - G) + \gamma\frac{900}{T+273}u_2(e_a - e_0)}{\Delta + \gamma(1 + 0.34u_2)} \tag{1}$$

where it is assumed that the height of the reference crop is 0.12 m, the surface impedance is 70 s/m, and the albedo is 0.23 [56]. The $ET_0$ represents reference crop evapotranspiration (mm·day$^{-1}$), $\Delta$ represents the slope of saturated vapor pressure (kPa·$°$C$^{-1}$), $R_n$ represents the net radiation (MJ·m$^{-2}$·d$^{-1}$), $G$ represents the soil heat flux (MJ·m$^{-2}$·d$^{-1}$), $e_a$ is the saturated vapor pressure (kPa), $e_0$ is the actual water vapor pressure (kPa), $\gamma$ is the hygrometer constant (kPa·$°$C$^{-1}$), $T$ is the air temperature at 2 m ($°$C), and $u_2$ is the wind speed at 2 m (m·s$^{-1}$).

As digital elevation model (DEM), slope and aspect vary little over time, the data are all based on one image from the same time period (https://www.gscloud.cn, accessed on 7 February 2023). Finally, the temporal and spatial resolution of all products used in this study were standardized to 1-day/1 km before using the constructed model to predict the regional LE.

## 3. Method

### 3.1. Transfer Learning Models

In this study, five machine learning algorithms, including ANN, RF, SVM, XGBoost, and LightGBM, were used to build transfer learning models. In the machine learning methods, RF builds multiple independent regression trees and merges them together to obtain more accurate and stable predictions. SVM achieves linear regression by projecting the original finite dimensional dataset into a higher dimensional space using a kernel function, which can achieve relatively high accuracy and generalization ability when the sample size is not large enough or the dimensionality of the features is greater than the number of samples. ANN continuously adjusts the connection weights of nodes in the hidden layers to obtain the best model through a large amount of data, and has strong fault tolerance and learning ability. The type of neural network used in this study is back propagation (BP) [57] with one input layer, three hidden layers and one output layer. XGBoost is a decision tree algorithm based on the gradient boosting framework and possesses the capability to prevent overfitting issues. Similarly, LightGBM is another framework for implementing the gradient boosting decision tree (GBDT) algorithm, but it employs a histogram-based decision tree algorithm instead.

To maximize the extraction of similar data, five driving variables under four different land cover types were selected, which have the highest Pearson product-moment correlation coefficient (PPMCC) with LE (see details in Table 5). Regarding instance-transfer, it is important that there is some similarity between the transferred data and the source data to avoid interfering with the input dataset and leading to a reduction in model accuracy. Therefore, the five driving variables and LE in each record of the observed data were converted into vector and the mean of the vectors was calculated for the different land cover types in HRB. Then the difference between the vector and the mean vector was obtained for each external data to determine the similarity of each external data to the source data. In this study, we extracted similar external data under each land cover class in increasing order of the normalized Euclidean distance (NED) to form a new dataset, which is consistent with the size of the HRB dataset.

**Table 5.** PPMCC between driving variables and latent heat flux for primary land cover types.

| Variable | BSV | CRO | CSH | GRA |
|---|---|---|---|---|
| Ta | 0.3574 | **0.7303** | **0.7489** | **0.6702** |
| RH | **0.2683** | −0.0434 | −0.1845 | 0.1553 |
| P | 0.2690 | 0.4170 | 0.0651 | 0.2341 |
| Rn | 0.4511 | **0.8280** | **0.6907** | **0.7270** |
| SM | **0.5056** | 0.4879 | 0.1659 | 0.4206 |
| WS | −0.0642 | −0.0359 | −0.0153 | −0.0862 |
| VPD | 0.1525 | 0.5953 | **0.7589** | 0.4242 |
| Pa | −0.2647 | −0.2358 | −0.2081 | 0.0460 |
| $ET_0$ | **0.4073** | **0.8280** | 0.5333 | **0.7550** |
| ASPECT | −0.2362 | 0.1563 | 0.0300 | 0.0320 |
| DEM | 0.1491 | 0.2238 | 0.0300 | −0.0223 |
| SLOPE | 0.1723 | 0.2288 | 0.0300 | −0.0474 |
| NDVI | **0.4204** | **0.7486** | **0.7846** | **0.6044** |
| LAI | **0.4555** | **0.6448** | **0.8075** | **0.4996** |
| DOY | 0.1275 | 0.0507 | 0.0614 | 0.0160 |
| month | 0.1277 | 0.0511 | 0.0632 | 0.0181 |

The five driving variables with the highest PPMCC are marked in bold for each land cover type.

In this study, two different data selection scenarios were developed to extract similar external datasets of HRB. In the first scenario, data were randomly selected from similar external data by increasing 10% each time from 0% to 100% to form the training set of the model with the current dataset; in the second scenario, data were also selected at a scale of 0–100% but at NED increment. Subsequently, 11 experiments were conducted on the ANN model using two distinct methods. The root mean squared errors (RMSE) and coefficient of determination ($R^2$) were selected to determine the ratio of Heihe River data to similar external data.

### 3.2. BTCH Method

The main principle of the BTCH method is to calculate the relative uncertainty of each group of LE by the three-cornered hat (TCH) method and derives the corresponding weights, and finally to obtain the fused LE based on the weights [58]. The probability density for the scale-expansion results of the transfer learning model is calculated as,

$$p(LE_i|LE_t) = \frac{1}{\sigma_i\sqrt{2\pi}}exp\left[-\frac{\varepsilon_i^2}{2\sigma_i^2}\right] = L(LE_t|LE_i)(\varepsilon_i = LE_i - LE_t) \qquad (2)$$

where $i$ is the $i$th model, $LE_t$ is the true value of LE at day $t$, $\varepsilon_i$ and $\sigma_i$ are the zero-mean white noise and error variance of the $i$th LE, respectively, and $L(\cdot)$ is the likelihood function.

Similarly, the probability density function for the results of the $j$ th LE simulation is as,

$$p(LE_j|LE_t) = \frac{1}{\sigma_j\sqrt{2\pi}}exp\left[-\frac{\varepsilon_j^2}{2\sigma_j^2}\right] = L(LE_t|LE_j)(\varepsilon_j = LE_j - LE_t) \qquad (3)$$

where $\varepsilon_j$ and $\sigma_j$ are the zero-mean white noise and error variance of the j th LE.

The maximum likelihood of the true value of LE ($LE_t$) is the maximum of its joint probability distribution,

$$maxL(LE_t|LE_i, LE_j) = p(LE_i|LE_t)p(LE_j|LE_t) = \frac{1}{2\pi\sigma_i\sigma_j}exp\left[-\frac{\varepsilon_i^2}{2\sigma_i^2} - \frac{\varepsilon_j^2}{2\sigma_j^2}\right] \qquad (4)$$

$LE_t$ can be obtained by defining the cost function and setting the first variable to zero:

$$LE_t = \frac{\sigma_i^2}{\sigma_i^2 + \sigma_j^2} LE_i + \frac{\sigma_j^2}{\sigma_i^2 + \sigma_j^2} LE_j \tag{5}$$

We define $LE_t$ such that $LE_t = w_i LE_i + w_j LE_j$ where $w_i$ and $w_j$ are defined as follows:

$$wi = \frac{\sigma_i^2}{\sigma_i^2 + \sigma_j^2}, wj = \frac{\sigma_j^2}{\sigma_i^2 + \sigma_j^2} \tag{6}$$

The weight of each group of LE simulation results (e.g., $w_i$) can be obtained by minimizing the similarity cost function:

$$w_k = \frac{\prod_{i=1, i \neq k}^{N} \sigma_i^2}{\sum_{k=1}^{N} \left( \prod_{i=1, i \neq k}^{N} \sigma_i^2 \right)}, \tag{7}$$

where the error covariance ($\sigma_i$) of LE can be obtained using the TCH method [59,60].

### 3.3. Evaluation Metrics

We used four metrics to evaluate the data selection and the model performance, including *PPMCC*, *NED*, $R^2$, and *RMSE*. The four formulas are expressed as follows:

$$PPMCC = \frac{\sum_{i=1}^{N} (X_i - \bar{X})(Y_i - \bar{Y})}{\sqrt{\sum_{i=1}^{N} (X_i - \bar{X})^2} \sqrt{\sum_{i=1}^{N} (Y_i - \bar{Y})^2}} \tag{8}$$

$$NED = \sqrt{\sum_{i=1}^{N} \left( \frac{X_i - Y_i}{S_i} \right)^2} \tag{9}$$

$$R^2 = \frac{\sum_{i=1}^{N} (\hat{y_i} - \bar{y})^2}{\sum_{i=1}^{N} (y_i - \bar{y})^2} \tag{10}$$

$$RMSE = \sqrt{\frac{\sum_{i=1}^{N} \left( y_i - \hat{y_i} \right)^2}{N}} \tag{11}$$

where $X_i$ and $Y_i$ represent the two sets of data to be compared, and $S_i$ is the variance between them; $\bar{X}$ and $\bar{Y}$ represent the mean value of the two sets of data, respectively; $y_i$ is the observed value, $\hat{y_i}$ is the predicted value, and $\bar{y}$ is the mean value of observations; $N$ is the number of total samples.

### 3.4. The Upscaling Model Framework

The procedure of the upscaling model can be illustrated in Figure 2. The model uses site data in HRB and similar external data as inputs. In order to achieve greater accuracy in obtaining the transfer learning models, the screening of variables and the determination of transfer ratio were carried out separately. Afterwards, based on the new dataset (as shown in Figure 2a), five upscaling models were implemented. In addition, Figure 2b illustrates the process of acquiring regional-scale LE in HRB. For further information about the transfer learning model setup and BTCH algorithm, see Sections 3.1 and 3.2, respectively.

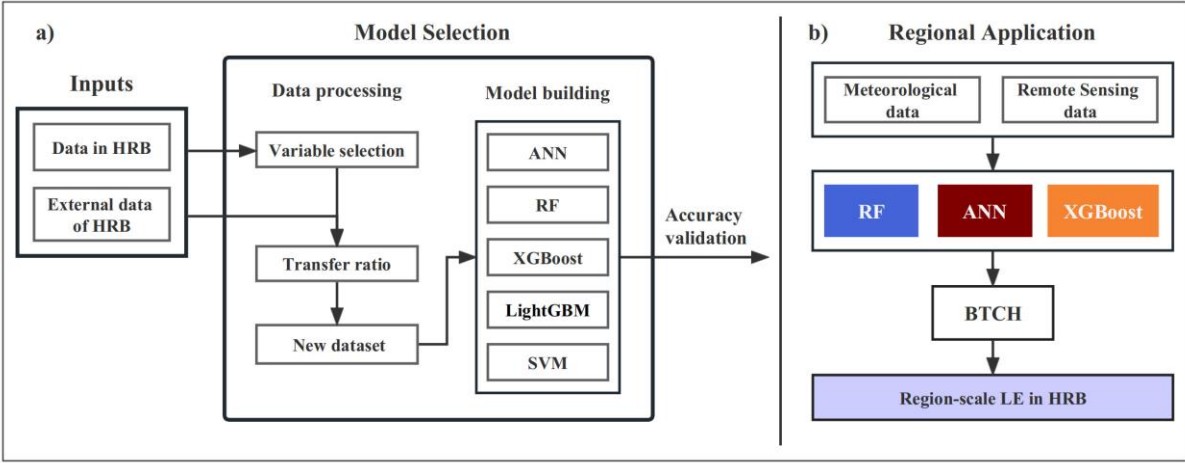

**Figure 2.** The latent heat flux upscaling model framework. (**a**) Building and screening of transfer learning models. (**b**) Regional application of the fusion model based on the BTCH method.

## 4. Results and Discussion

### 4.1. Effectiveness of External Data for Upscaling Model

Considering that LE is influenced by a variety of factors, in order to obtain the best performance of transfer learning model, PPMCC was used to evaluate the similarity of the driving variables under four different land cover types, which are BSV, CRO, CSH, and GRA, respectively. Table 5 shows a strong correlation between Rn, $ET_0$, and LE under all four land cover types, which is consistent with the physical mechanism in the P-M equation. The correlations between LE and vegetation factors, as well as Ta, were found to be stronger at sites with high vegetation cover compared to those at BSV sites. In contrast, the correlation with SM was more significant at the BSV sites than CRO sites, due to the fact that vegetation transpiration can utilize deeper soil moisture. Finally, for each land cover type, the five driving variables (bolded) with the highest correlation coefficients were selected.

On the basis of the five variables, data were introduced from similar external data using the two scenarios already mentioned. The difference between the two scenarios is that the second ensures that the data extracted every time is the most similar to the HRB dataset. Figure 3 shows that the second scenario performs better overall than the former. For the first one, the inclusion of data resulted in a flatter change in ANN model performance. When the incorporation ratio (the data samples ratio between HRB and external dataset) is 5:3 (transfer ratio is 0.6), the model achieves the best RMSE and $R^2$ in both cases, but the second has a lower RMSE and a higher $R^2$. Therefore, the model was constructed by introducing external data according to the law of increasing NED.

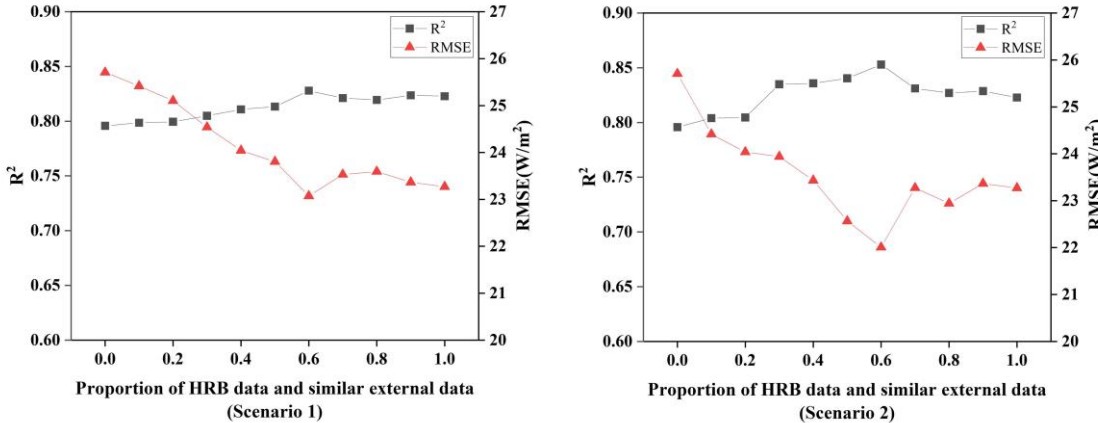

**Figure 3.** Performance of ANN models in two scenarios.

When calculating the proportion of external data introduced, the ANN model performs best when data sample transfer ratio is 0.6 (RMSE = 22.00 W/m$^2$, R$^2$ = 0.85), comparing with the original ANN results without external data (RMSE = 25.71 W/m$^2$, R$^2$ = 0.80). From this point, the performance on the test set decreases slightly before settling into a range that is still better than when no similar external data is added. It also suggests that the training set plays a role in the final performance of the machine learning model, with the more comprehensive the data in the training set, the better the generalization ability of the model. When the ratio exceeded 0.6, the model did not work well instead, probably because the ratio of similar external data was too large and the model did not capture enough key features of the data in HRB, which eventually led to some errors in the test set.

### 4.2. Comparison of the Results from Five Transfer Learning Models

In this study, the performances of the five transfer learning models were evaluated and analyzed using the hold-out method, where the ratio of the training set to the test set is 8:2. The scatterplots of predicted LE versus observed LE for the five models over the test set were plotted in Figure 4. As shown, ANN and RF algorithms outperformed the other models significantly. The ANN model has the largest R$^2$ of 0.85 and the RF model has the lowest RMSE of 20.97 W/m$^2$. Compared to the RF model, the slope of the fitted line for the ANN model is closer to 1, but when some of the data in the training set is small (LE < 200 W/m$^2$), the ANN model has a simulated value of 400–600 W/m$^2$, which results in a larger RMSE for the ANN model than the RF model. As the SVM model is essentially a linear simulation with the least effective, the scatter points of models other than itself are generally distributed around the 1:1 line. A common problem with the XGBoost and LightGBM models, however, is that for models trained on a training set with no negative values, negative values exist in the predictions. Non-negative site observations of LE values can be predicted to a maximum of around −20 W/m$^2$, especially in the LightGBM model.

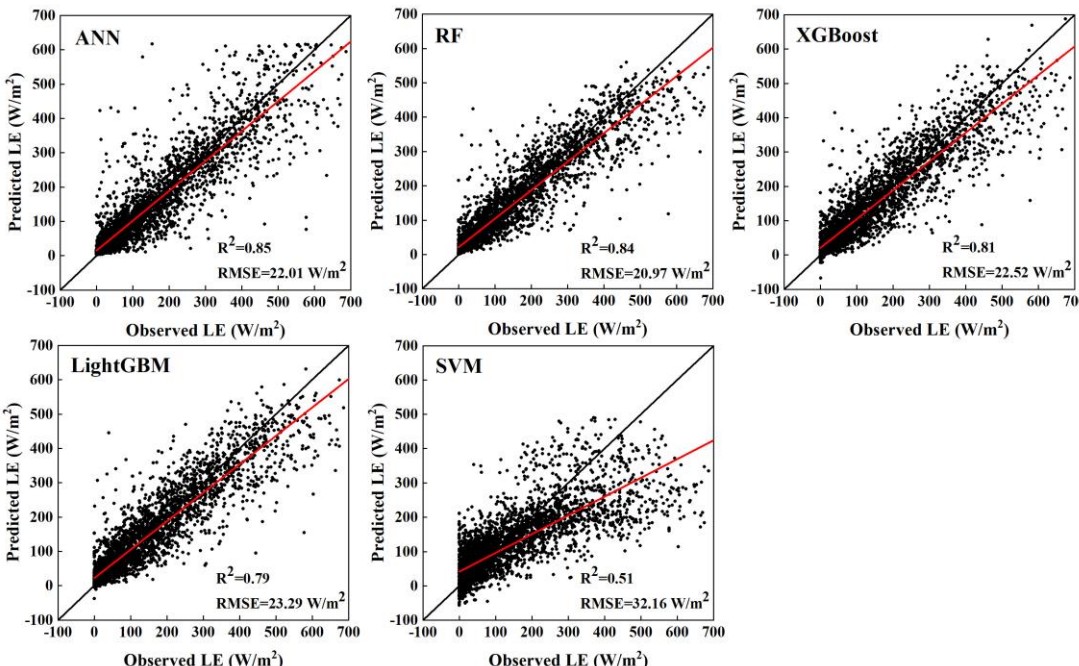

**Figure 4.** Scatterplots of the five transfer learning models over the test set.

Based on the results of the five models, it can be found that the models cannot perform well when the LE observations are at higher or lower values. The main reason for this phenomenon may be that evaporation and transpiration require a certain time process, and there may be a certain lag in the relationship between LE and precipitation and soil moisture.

### 4.3. Accuracy Validation and Time Series Analysis

The best three models, ANN, RF, and XGBoost, were used to produce 1 km×1 km daily LE over HRB throughout 2018, using meteorological and remote sensing forcing data as input. The results of the three models were then fused using the BTCH method to obtain a new set of daily maps of LE. The spatial distribution of the fusion result can be found in Figure 5.

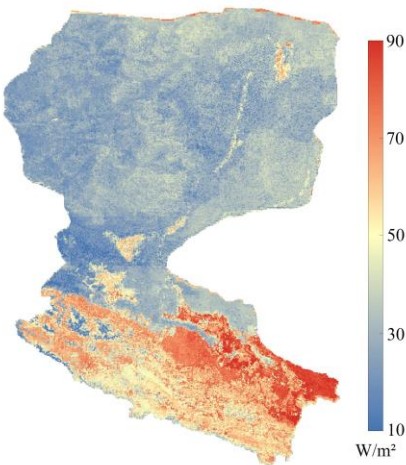

**Figure 5.** Spatial distribution of annual average latent heat flux over HRB in 2018.

The daily LE obtained from the three single transfer learning models and their fusion were compared with LAS observations in HRB. Results (Table 6) show that the calculation accuracies of the Arou (RMSE = 19.80 W/m$^2$, R$^2$ = 0.80), Daman (RMSE = 21.48 W/m$^2$, R$^2$ = 0.77), and Sidaoqiao (RMSE = 22.69 W/m$^2$, R$^2$ = 0.76) sites are consistent with the heterogeneity of the underlying surface at the upper, middle and lower reaches of HRB sites. The underlying surface of the Arou station is relatively homogeneous and the main land cover type is grassland. For Daman station, there are roads, villages, and orchards in addition to maize farmland. The underlying surface of the Sidaoqiao station is the most complex and fragmented, with high spatial heterogeneity, and the land cover types include tamarisk, Populus euphratica, crop land, desert, etc. For model algorithms, RF and ANN are significantly better than XGBoost, while the XGBoost model has the worst performance. The BTCH model has the best simulation result, with R$^2$ and RMSE better than others. The fusion approach retains the advantages of three transfer learning models and also provides some improvement to the phenomenon of overestimation of low values at Daman station.

**Table 6.** Accuracy validation of latent heat flux simulated at upstream, midstream, and downstream LAS sites.

| Site Name | Arou | | Daman | | Sidaoqiao | | Mean | |
|---|---|---|---|---|---|---|---|---|
| | RMSE (W/m$^2$) | R$^2$ | RMSE (W/m$^2$) | R$^2$ | RMSE (W/m$^2$) | R$^2$ | RMSE (W/m$^2$) | R$^2$ |
| ANN | 20.30 | 0.79 | 19.90 | 0.80 | 22.26 | 0.76 | 20.82 | 0.78 |
| RF | 18.37 | 0.83 | 21.13 | 0.78 | 23.32 | 0.75 | 20.94 | 0.79 |
| XGBoost | 23.32 | 0.73 | 26.41 | 0.67 | 24.36 | 0.72 | 24.70 | 0.71 |
| BTCH | 17.23 | 0.85 | 18.48 | 0.83 | 20.80 | 0.80 | 18.84 | 0.83 |

The discrepancies between transfer learning models and fusion model were mainly caused by model algorithms, observation data uncertainty, limitations of the training set, and land surface heterogeneity. Firstly, machine learning algorithms are essentially learning about data and the underlying relationships among data through continuous iterative learning of the training set. The differences in model principles lead to different areas of expertise for each model, with the RF model performing the most consistently in

this study and the ANN model being more accurate in some extreme cases. The BTCH approach retains the strengths of three models through data fusion and the shortcomings of the models are compensated for. Secondly, the observed and inverse accuracy of the HRB data is not exactly the same as the true values, while differences in processing methods and different spatial and temporal resolutions can also introduce some errors. Thirdly, the training set is mainly derived from site observations, which are spread over a relatively concentrated farmlands and oases, so the model learns a limited amount of knowledge, probably causing some errors during the process of upscaling. Finally, LE is determined by numbers of factors, and verification accuracy is usually high over homogeneous stations (i.e., Arou) than heterogeneous stations (i.e., Sidaoqiao).

Moreover, this study also validated the temporal trends of LE simulated by the BTCH method using LAS observations (Figure 6). The simulation results for the upper, middle, and lower reaches are generally consistent with the LAS observations trends, especially when there were significant precipitation processes at the Arou station from DOY 125 to 154, the Daman station from DOY 175 to 200, and the Sidaoqiao station from DOY 225 to 250. However, mainly influenced by the worst performance of the XGBoost model at the Daman station (RMSE = 26.41 W/m$^2$, R$^2$ = 0.67), the time series fitting of the fusion result overestimated at the Daman station, especially when the values of LAS observations are low. Overall, the BTCH method combines the strengths of the three models and can simulate well when precipitation events occur.

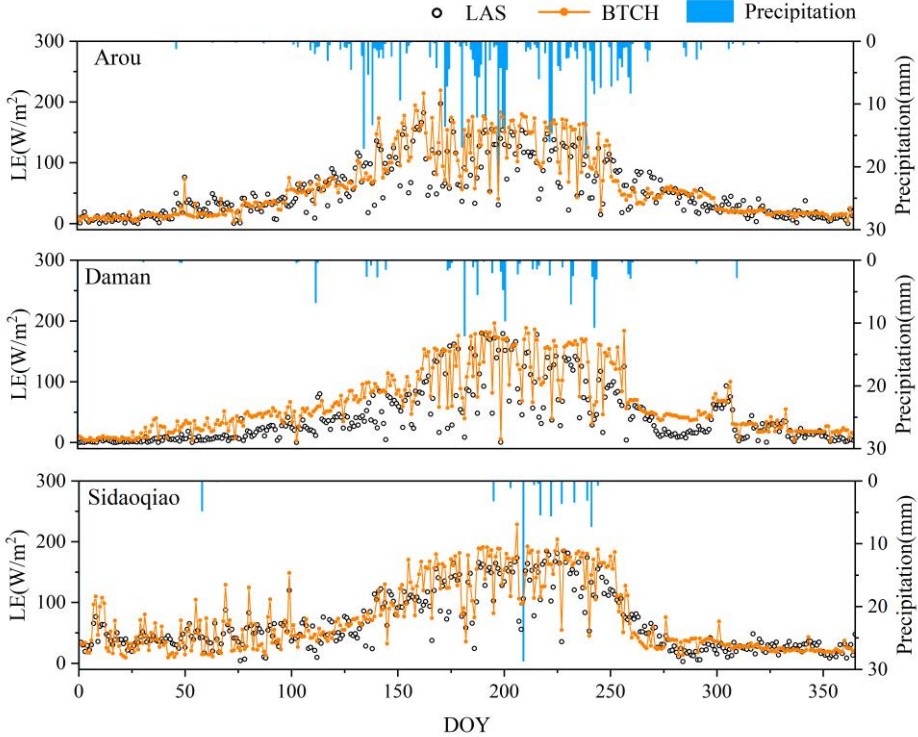

**Figure 6.** Temporal consistency of BTCH method with LAS observations.

### 4.4. Spatio-Temporal Characteristics of Fused Latent Heat Flux Upscaling Results

To investigate the value of the fused LE, we explored the spatial and temporal distribution of LE in HRB during the growing season (May to October) in 2018 (Figure 7). In terms of time scale, LE over all land cover types is greatest in July or August, with a decreasing trend in the upstream to downstream direction, which is also consistent with site observations. The LE in the oasis zone of the basin rises continuously from May to August and decreases from August to October mainly due to the influence of the temperate continental monsoon climate, which is also verified by the precipitation histograms. At the spatial scale, the maximum values of LE are found in the woodlands and grasslands of the

upper reaches, the farmlands of the middle reaches, the oases of the lower reaches, and the area around the HRB, while the minimum values are mainly in the desert and bare land of the lower reaches. The main reason is the difference in vegetation cover and wetness of the underlying surface in the basin, where transpiration and surface evaporation are stronger in densely vegetated areas resulting in higher LE. The LE is significantly higher in the farmlands (mostly maize) of the midstream than in other land cover types, mainly owing to the higher transpiration capacity of the crops and artificial irrigation. Grassland, woodland, and scrub have lower evapotranspiration than the above vegetation types, which is related to the relatively weak transpiration capacity of grassland and the low cover of woodland. In addition, evapotranspiration is also very high in the wetland due to the abundant natural water supply. In addition, the LE in the desert bare ground is generally lower in the downstream than in the midstream.

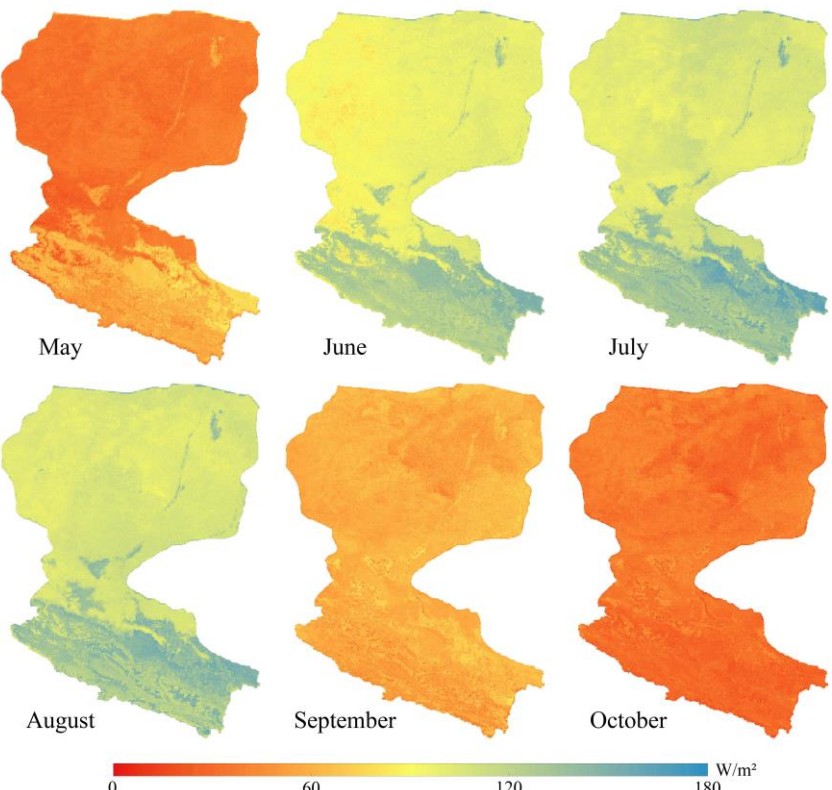

**Figure 7.** Spatial and temporal distribution of latent heat flux in HRB during the growing season (May to October) in 2018.

In addition, Ta, Rn, NDVI, and LAI were selected to calculate the PPMCC with the fusion result separately to analyze the reasonableness of the spatial and temporal trends. Figure 8 shows that Ta and Rn have good positive correlations with LE throughout the eastern part of HRB, especially in the oases of the upstream and the farmland of midstream, where the PPMCC is close to 1. However, there is a clear boundary between the eastern and western parts of the downstream in the two plots, with west of the boundary the PPMCC between Ta/Rn and LE decreasing and then showing a strong negative correlation. The main reason is that the majority of the sites selected for the study are located east of the boundary, with a scarcity of sites in the west. In farmland, NDVI and LAI have a very strong positive correlation with LE, due to the fact that there is an adequate supply of water in irrigated farmland, making LE more controlled by vegetation growth and energy factors. However, in the desert areas of the downstream, the spatial distributions of NDVI and LE correlation are less regular with many zero values, probably due to some errors introduced during interpolation of NDVI data. Furthermore, the PPMCC could not be calculated due to the missing LAI data in these areas for a long time.

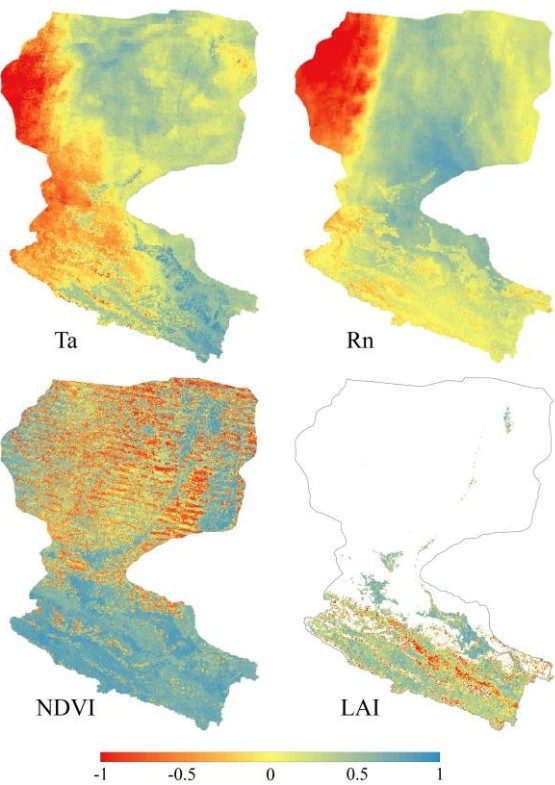

**Figure 8.** Spatial distribution of the PPMCC between environmental variables and latent heat flux.

## 5. Conclusions

In order to upscale LE from site scale to regional scale with the spatial resolution of 1 km in HRB, this study introduced five transfer learning models and the BTCH fusion method. Specifically, five machine learning algorithms were utilized to construct the upscaling framework, namely ANN, RF, SVM, XGBoost, and LightGBM. Then, the three best upscaled results were selected to fuse based on the BTCH method. Finally, the upscaled results before and after fusion were validated against the LAS observations, and the main conclusions are as follows.

Introducing the idea of instance transfer to upscale LE has improved the accuracy of the machine learning models to some extent. By selecting proper variables from external datasets under different land cover types and establishing a transfer ratio of 0.6 between external and internal datasets, the upscaling model can grasp more effective information, resulting in an improvement or reduction of 6% and 17% in the $R^2$ and RMSE values, respectively, for the ANN model. Among the five models, ANN, RF, and XGBoost models are the most suitable to upscale LE from site scale to regional scale in HRB, with the best fit of the simulated values to the site observations, the smallest errors, and less outliers.

Generally speaking, different algorithms utilized to develop upscaling models may show some differences in temporal and spatial distribution. To integrate their strengths, this study applied the BTCH fusion approach to obtain the final upscaling LE product in HRB. The verification results with precipitation and observed LE indicate that the fusion dataset retains the advantages of the different results, with reasonable time trends and the highest accuracy ($R^2 = 0.83$, RMSE = 18.84 W/m$^2$). Moreover, the fusion results also show that the maximum values of LE on a temporal scale occur in July and August, with increasing from May to August and decreasing from August to October in the oases of upper and middle-lower reaches. From the perspective of spatial scale, the maximum values of LE are found in the upstream woodlands and grasslands, the midstream farmlands, the downstream oases, and the area around the HRB. However, the minimum values are mainly in the downstream desert and bare ground, with a decreasing trend in the upstream to downstream direction.

**Author Contributions:** Conceptualization, T.X.; methodology, X.H., T.X. and J.L.; formal analysis, X.H.; writing—original draft preparation, J.L.; writing—review and editing, All; visualization, X.H. and J.L.; supervision, T.X., S.L., Z.X. (Ziwei Xu) and L.Z.; project administration, T.X.; funding acquisition, T.X. All authors have read and agreed to the published version of the manuscript.

**Funding:** This work was funded by National Key R&D Program of China (Grant No. 2021YFB3900505), the National Natural Science Foundation of China (42171315) and the State Key Laboratory of Earth Surface Processes and Resource Ecology (2021-ZD-04).

**Data Availability Statement:** The site observations around HRB, meteorological forcing data, land surface soil moisture dataset of Qilian Mountain Area and Rn dataset in HRB were all downloaded from National Tibetan Plateau Data Center (https://data.tpdc.ac.cn/home, accessed on 1 February 2023). Some of the similar external datasets were downloaded from FLUXNET 2015 (https://fluxnet.org/data/fluxnet2015-dataset, accessed on 5 February 2023).The remote sensing datasets including LAI, NDVI, and LC were downloaded from: https://modis.gsfc.nasa.gov (accessed on 7 February 2023). The DEM, SLOPE, and ASPECT data can be downloaded from https://www.gscloud.cn (accessed on 7 February 2023).

**Conflicts of Interest:** The authors declare no conflict of interest.

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
