# Peer review of "Upscaling of Latent Heat Flux in Heihe River Basin Based on Transfer Learning Model"

_remotesensing, doi:10.3390/rs15071901_

Round 1

Reviewer 1 Report

This is an interesting topic to improve latent heat flux estimation in the Heihe River basin based on the transfer learning model. In addition, BTCH is an innovative approach to merge three pre-selected machine learning LE estimates. The paper is readable and easy to follow. Overall, I think the current work well presented, the results nicely summarized and convincingly. I only have some minor comments on the current manuscript for the attention of the authors. 

Line 99-109: This paragraph needs to start by emphasizing the significance and function of the BTCH approach.

Figure 1: The red rectangular area is not consistent with the amplified area.

Line 141: The term "near the oases" is inaccurate because most of the sites are established in the oases.

Line 200: The reference crop evapotranspiration is not list in Table 4.

Line 232-234: Which five driving variables are used in the machine learning approach? it is unclear.

Figure 2b: In this framework, the three machine learning methods are unknown because you have not compared and validated them.

Figure 6: The BTCH method systematically overestimates LE compared to observations. Does this mean that all three methods being fused are systematically biased? Systematic bias is not only affected by the xgboost method (Line 403-405).

Line 450, use of ‘environmental variables’ is better than ‘influence factors’

Reviewer 2 Report

Q1: Table 4, please split the time resolution and space resolution in two columns.

Q2: It is not clear from the introduction why the Bayesian-based three-cornered hat method is chosen to fuse the upscaled results.

Q3: Line 212 and 213, the expression is not clear enough. It is recommended to add 'before using the constructed model to predict the regional LE' at the end of the sentence to better distinguish the datasets used during model construction and application.

Q4: Why is the variable ET0 chosen when selecting driving variables for the upscaling model, and what impact does its addition have on the machine learning model.

Q5: Please ensure consistency in the decimal places used for latitude and longitude across Tables 1, 2, and 3.

Q6: The caption of Table 5 is not accurately presented. It is recommended to revise it as 'PPMCC between driving variables and latent heat flux for primary land cover types.

Q7: Figure 5, 7, and 8, please incorporate a specific number of additional annotations into the colorbars to enhance the clarity of the results. Adding these annotations will provide viewers with a more detailed understanding of the data.
